# Development of a competency profile for professionals involved in infectious disease preparedness and response in the air transport public health sector

Doret de Rooij[1,2]*, Rebekka Rebel[1,2], Jörg Raab[3], Christos Hadjichristodoulou[4], Evelien Belfroid[1], Aura Timen[1,2]

1 Centre for Infectious Disease Control, National Institute for Public Health and the Environment, Bilthoven, The Netherlands, 2 Athena Institute, Free University, Amsterdam, The Netherlands, 3 Department of Organization Studies, School of Social and Behavioral Sciences, Tilburg University, Tilburg, The Netherlands, 4 Department of Hygiene and Epidemiology, University of Thessaly, Thessaly, Greece

* doret.de.rooij@rivm.nl

**Data Availability Statement:** All relevant data are within the paper and its Supporting Information files.

## Abstract

### Background

Recent infectious disease outbreaks highlight the importance of competent professionals with expertise on public health preparedness and response at airports. The availability of a competency profile for this workforce supports efficient education and training. Although competency profiles for infectious disease control professionals are available, none addresses the complex airport environment. Therefore, the main aim of this study is to develop a competency profile for professionals involved in infectious disease preparedness and response at airports in order to stimulate and direct further education and training.

### Methods

We developed the competency profile through the following steps: 1) extraction of competencies from relevant literature, 2) assessment of the profile in a national RAND modified Delphi study with an interdisciplinary expert group (n = 9) and 3) assessment of the profile in an international RAND modified Delphi study with an airport infectious disease management panel of ten European countries (n = 10).

### Results

We systematically studied two competency profiles on infectious disease control and three air transport guidelines on event management, and extracted 61 relevant competencies for airports. The two RAND modified Delphi procedures further refined the profile, mainly by specifying a competency's target group, the organizational level it should be present on, and the exact actions that should be mastered. The final profile, consisting of 59 competencies, covers the whole process from infectious disease preparedness, through the response phase and the recovery at airports.

**Funding:** This publication has been produced with the support of the European Commission's Consumers, Health, Agriculture and Food Executive Agency (CHAFEA) for the Healthy Gateways Joint Action (grant agreement no. 801493) and support from the Dutch Ministry of Health, Welfare and Sport. The content represents the views of the author only and is his/her sole responsibility; it cannot be considered to reflect the views of the European Commission and/or the Consumers, Health, Agriculture and Food Executive Agency (CHAFEA) or any other body of the European Union. The European Commission and the Agency do not accept any responsibility for use that may be made of the information it contains.

**Competing interests:** The authors have declared that no competing interests exist.

## Conclusion

We designed a profile to support training and exercising the multidisciplinary group of professionals in infectious disease management in the airport setting, and which is ready for use in practice. The many adaptations and adjustments that were needed to develop this profile out of existing profiles and air transport guidelines suggest that other setting-specific profiles in infectious disease control are desirable.

## Introduction

Nowadays, people travel further, faster, and in higher numbers than ever before [1]. While the increased speed of human mobility has economic advantages [2], it also has created a global transport network for infectious diseases [3, 4]. Air travel and airports contribute to this network due to the high number of passengers from across the world coming together in confined spaces. One contagious passenger can spread the illness to numerous other passengers, crew, and airport personnel [5]. The volume of air travel continues to increase annually, as well as the amount of infections imported by travelers [6, 7]. It is, therefore, crucial that airports have enough capacity to prevent the rapid spread of infectious diseases to other areas [8–10]. According to the International Health Regulations (IHR), countries have to designate their major points of entry and to assure that these have implemented the core capacities required for infectious disease management [9].

As is revealed by previous studies, strong capacity alone is not sufficient. During outbreaks of Severe Acute Respiratory Syndrome (SARS), Ebola Virus Disease (EVD), and Zika Virus disease, there was a lack of clarity on responsibilities among airport personnel and public health authorities [5], leading to insufficient communication and coordination among local and national stakeholders [3, 5]. In the case of SARS, the delay in the implementation of control measures tripled the epidemic size of the outbreak [11]. While surveillance systems recognize the event, we need policies and potential measures in place and professionals able to act upon them [11–13]. While capacities are system-level characteristics, these are enforced by teams of individual, knowledgeable, skilled and dedicated professionals, i.e. capacity is enforced by competency [14].

Professionals are competent if they demonstrate the attitude, knowledge and skills required in their particular profession [13, 15]. Well-formulated competencies define what is required of professionals [16, 17], and provide a foundation to build an effective and targeted training [18]. Competency-based learning and training are widely integrated into medical education [19,20], and the number of competency profiles for health care professionals, including those for infectious disease control, continue to grow [21–29]. However, responding to infectious diseases in the complex air travel environment requires specific competencies on top of that. Professionals need to perform an appropriate public health risk assessment, manage cases, collect passenger information, notify national authorities, and communicate to passengers and multiple stakeholders [29]. Furthermore, all these measures should be proportionate to the risk without any interference with the international travel and trade [9]. The public health authority and relevant airport operators are mainly responsible for infectious disease control at the airport [30]. Nonetheless, many other entities are also involved and need to coordinate effectively with the public health authority. Developing a competency profile addressing only one particular discipline is therefore impracticable for the air travel environment.

To the best of our knowledge, there is currently no profile that describes the competencies that professionals involved in infectious disease management at airports need. Therefore, this study aimed to develop such a profile that describes the competencies that a multidisciplinary group need–in addition to the basic knowledge and skills required for their profession–when called upon to prepare and respond to infectious diseases in the airport environment.

## Methodology

### Study design

We performed the systematic RAND modified Delphi consensus procedure [31] to develop a competency profile for professionals involved in infectious disease preparedness and response at airports. Competencies were extracted from existing competency profiles and air transport guidelines on event management and consecutively presented to a national and an international panel of professionals. Both panels followed a methodologically identical path (**Fig 1**) that consisted of two rounds of data collection. First, panels assessed the relevance of competencies regarding their expertise during a digital questionnaire. Next, consensus meetings were held to gather consensus on undecided competencies. The refined final competency profile based on the first national round formed the concept competency profile for the international validation. This method is commonly used for the development of competency profiles [32], using the collective subjective judgment of professionals [33].

Step 2 and 3 were performed twice in this study: first with a national expert panel and after with an international expert panel.

### Study population and recruitment

The study population of the national panel consisted of professionals involved in infectious disease preparedness and response at Schiphol Airport. This airport is the only IHR designated airport in the Netherlands, one of Europe's largest airports and a major hub in international

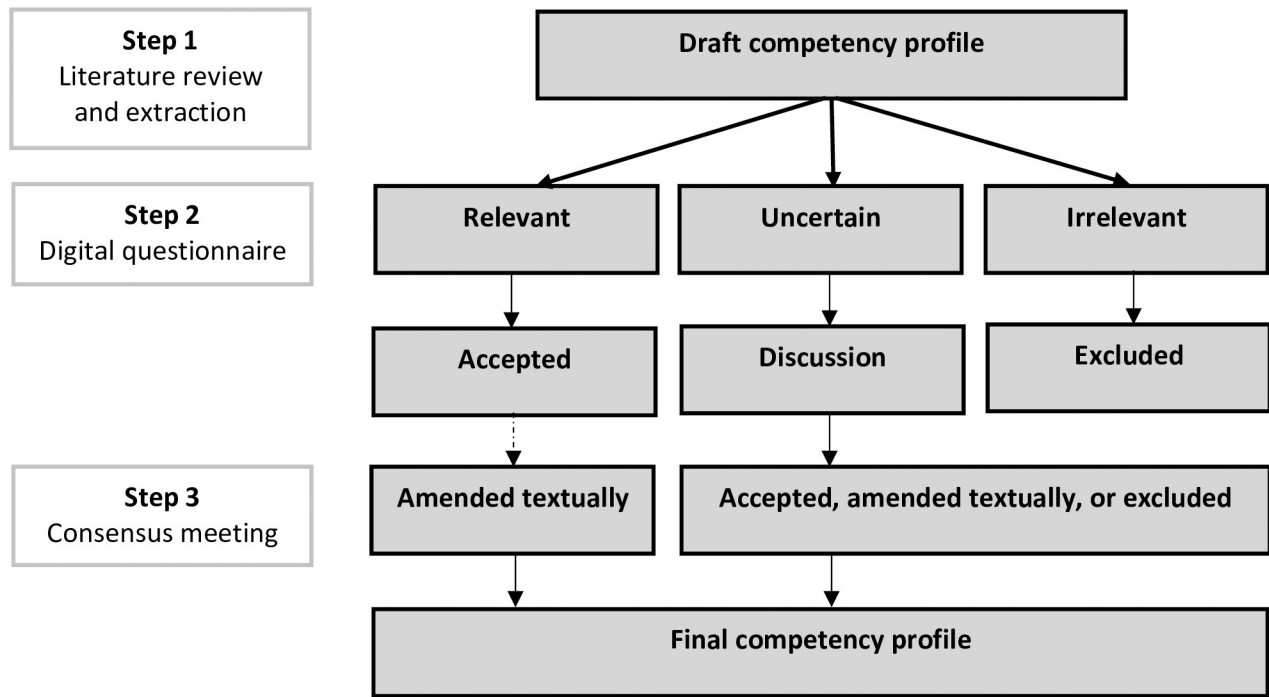

**Fig 1. The application of the RAND modified Delphi procedure.**

air traffic [34]. The study population of the international panel consisted of professionals involved at a local or national level in infectious disease preparedness and response at major European airports. The research team aimed for two panels consisting out of 8–15 participants [31], resulting in a national panel of 9 and international panel of 10 participants. In this way, we tried to assure both enough reflection of different opinions and a setting for a safe and fruitful discussion. The panels were recruited using purposive sampling according to inclusion criteria until a representative panel was composed or time was limited for further inclusions. The research team composed an invitation letter by e-mail explaining the background, aim, and method of the study for both panels, through which professionals could confirm their participation.

For the national panel, we consulted an experienced nurse from the public health service at Schiphol Airport to discuss our sample. We invited experienced professionals for the national round from the following disciplines: International Medical Advice, Airport Medical Services, public health service, Air Traffic Control, Flow Management of Aircrafts and passengers, Airport fire officer, Continuity & Crisis Management, and the Disaster Medicine Organization. International Medical Advice, Airport Medical Services, and the Public Health Service collaboratively ensure that the infectious disease policy is feasible and effective at the airport. The Public Health Service is responsible for the public health response. At their request, International Medical Advice facilitates medical items on board, such as personal protective equipment, and debrief crew and ground staff of an affected aircraft. In the situation that medical care is required, the ambulance will assist passengers upon arrival. Professionals from Air Traffic Control receive warnings from the pilot in command of a possibly affected aircraft and are responsible for informing the Flow Management of Aircrafts and passengers which adjusts logistical processes at the airport. The Airport Fire Officer ensures coordination among various emergency services during the management of an infectious disease case. The Continuity & Crisis Management is responsible for advising on risk, crisis, and physical safety management. Disaster Management Organization provides training in the field of infectious disease control at the airport.

The international panel was recruited by selecting professionals who participated in a European, face-to-face 3-day training course on infectious disease control at designated airports. This course was part of the European Union (EU) Joint Action Healthy Gateways [35] and took place on 18–20 September 2019 in Belgrade, Serbia. We used these professionals' self-declared competence on local and national level before the training to approach a variety of professionals. On the local level, a functional distinction was made between competence in preparedness planning, decision making, and implementation. We invited professionals in all competence areas and tried to include one participant per country, with countries geographically divided over Europe.

## Design

**Step 1 –Literature search and extraction of competencies.** We performed a literature search to identify competency profiles and air transport guidelines on event management. The search was executed in the 4th week of March 2019 using PubMed and Google Scholar. For competency profiles, search terms were 'infectious diseases' or 'public health', and 'competency' and their synonyms. For the airport guidelines, search terms were 'infectious diseases' or 'emergency preparedness' or 'public health event', together with 'airport' or 'air travel' or 'points of entry'. In Pubmed, the search terms were limited to the title or abstract. In Google Scholar, we searched several combinations of search terms and screened the results until 20

hits in a row were irrelevant. The search terms, the search strategy, the screening, and application of the selection criteria are shown in S1 File.

Regarding the competency profiles, the criteria for title/abstract screening were whether studies aimed at presenting competencies, and were aimed at infectious disease preparedness or control. Inclusion criteria for the full-text screening of the competency profiles were the explicit presentation of a competency set in the study, competencies aimed at graduate level or professionals in practice, applicable to infectious disease management. Further criteria to select the landmark studies contained a full array of competencies, being a peer-reviewed study, and being relevant for the airport setting.

Regarding the airport guidelines, the criteria for title/abstract screening were documents describing infectious disease management in the air travel setting with a disease-generic approach. The full-text screening inclusion criteria were documents aimed for the airport or air travel setting, prescribing infectious disease management in terms of specific preparedness and response tasks, with a disease a-specific approach. Further selection was made based on the target group at airport level, whether unique topics were prescribed compared with other included studies, and presenting a full array of the preparedness and response process.

The competency profiles and air transport guidelines were matched to develop a concept competency profile, using the following steps. First, we extracted main tasks for infectious disease preparedness and control from the guidelines that cover infectious disease management at airports during preparedness, response, and recovery. Then, the competency profiles and air transport guidelines were independently reviewed by two researchers (DdR & RR) to extract competencies or textual fragments that could be reformulated into a competency, and which were relevant for a task. Results were compared and dissimilarities were discussed until consensus was reached.

Subsequently, a data reduction round took place by applying three specifying criteria. First, we further narrowed the scope to the airport environment itself instead of competencies required regionally or countrywide. In addition, we restricted competencies on hygiene to the commanding level and excluded all operational cleaning. Lastly, competencies or textual fragments regarding general team competencies were excluded in the profile because the Team Resource Management (TRM) skill set is already available and includes these [36]. The TRM skill set, developed for aviation safety, covers skills such as interpersonal communication, leadership and decision making, and is complementary to our competency profile. Finally, we combined overlapping or congruent competencies and fragments. The included textual fragments were reformulated into competencies according to Bloom's Taxonomy of Education Objectives [37]. Data reduction was performed by RR & DdR collaboratively, all doubts and several versions of the draft profile were discussed with other researchers (EB and AT).

Next to specifying competencies to a task, we also specified them to a role to further enhance the logical structure and the usability of the profile. For this purpose, we adapted the CanMEDs Roles [38], which are widely used in medical education, into the following roles required in infectious disease management: (1) Health Expert, (2) Organizer (including policy development), (3) Scholar. Roles with respect to the Communicator, Professional and Collaborator are required in general, and are, therefore, considered general tasks. During several discussion meetings with a third researcher (AT), competencies were divided over tasks and roles and overlapping competencies were removed. This first step resulted in a concept competency profile.

**Step 2 –The national panel: Digital questionnaire & consensus meeting.** The next step contained the assessment of the draft competency profile in the national panel. A digital questionnaire was built in the web-based program Formdesk [39]. It was successfully pilot-tested on comprehensibility and usability by an experienced medical trainer working at the National

Institute for Public Health and the Environment, and a physician specialized in infectious diseases from the Maastricht Aachen Airport.

The digital questionnaire started with information regarding the aim and process of the study and key definitions. Also, participants received supportive documents on formulating competencies, according to Bloom's Taxonomy of Educational Objectives [37]. In addition, a privacy statement was included. The first questions were related to demographic characteristics, such as their gender, profession, the number of years in their current profession, and their experience with infectious disease preparedness and response at the airport on a 5-point Likert scale (1 = very inexperienced, 5 = very experienced). We requested e-mail addresses for planning the consensus meeting. Subsequently, the relevance of competencies in the concept competency profile was graded by the participants using the following question: 'To what extent do you consider this competency as a relevant element for infectious disease preparedness and response at airports'? Scoring was done using a 9-point Likert scale (1 = totally irrelevant, 9 = totally relevant). The introduction, privacy statement, and demographic questions were in Dutch, but the questions regarding the relevance of competencies, as well as the competencies were in English. All questions on relevance were mandatory. Participants could comment on individual competencies as well as suggest new ones per category. After grading each competency, participants were asked if the competency profile reflected their knowledge, skill, and attitude regarding infectious disease preparedness and response at the airport. Data were collected in the three weeks following 9 May 2019.

Participants who completed the digital questionnaire were invited to the consensus meeting. Participants received a personal feedback report in advance of the meeting. This report showed the group ratings of each competency, together with the participants' individual ratings. Participants could therefore identify major dissimilarities within ratings beforehand. The purpose of the meeting was to reach consensus by discussing these dissimilarities face to face. The moderator asked all participants to keep an incident in mind while considering the relevance of each competency. First, uncertain competencies were discussed, then competencies proposed by participants themselves, and lastly the suggestions for the reformulations of relevant competencies. If the participants could not reach consensus, a voting round was held to either accept, textually amend or reject the competency based on the majority opinion. After discussing all competencies, participants were asked whether the refined final competency profile reflected their knowledge, skills and attitude. Finally, the usability level of this profile was discussed, including any further suggestions for improvement. The national meeting was in Dutch because this was the native language of all participants. The meeting took place on 21 June 2019 at the training center of Schiphol Airport. An experienced moderator (AT) led the discussion.

**Step 3 –International validation: Digital questionnaire & consensus meeting.** The competency profile from step 2 functioned as the starting point for international validation. The competencies were already stated in English, but we had to translate the introduction, statements, and demographic questions into English. We asked the respondents in the introduction to keep a recent incident or outbreak in mind during the completion of the questionnaire and asked them to specify which event this was. We added numbers to the tasks to manage expectations on the size of the competency list during completion, and added a suggestion for additional competencies on data protection and privacy. Except from these changes, this international study was methodologically identical to the national study. The participants were invited on 12 August 2019 and were reminded twice by e-mail after two and four weeks. The data collection through the questionnaire ended on 13 September 2019. The consensus meeting took place on 19 September 2019 in Belgrade, Serbia, and was moderated by an experienced moderator.

## Data analysis

Data from Formdesk was transferred to an Excel file and checked on irregularities. First, a descriptive analysis of demographic characteristics was performed. Secondly, the median rating and the amount of dispersion of ratings between participants were calculated for each competency. If the competency had a median within 7–9 range and ≥70% of the participants scored it in the top tertile (score 7,8 or 9), then the competency was marked as 'accepted'. If the competency had a median of <7 and <70% scored in the top tertile, then the competency was marked as 'not accepted' and excluded. If the median score laid between 7–9 and <70% of the participants scored in the top tertile, then the competency was marked as 'uncertain'. In the national study, competencies with scores spread over all tertiles, despite any median, were also classified as 'uncertain'. Table 1 shows the classification of the competencies by the participants' median score and level of agreement. As the digital questionnaire contained open-ended questions and formulated competencies by the participants, responses were grouped and coded accordingly. To support the reliability, two researchers (DdR & RR) performed the data analysis independently and compared their results.

The consensus meetings were audio-taped and in outline transcribed. A distinction was made between individual opinions and contributions to the group consensus. In accordance with the General Data Protection Regulation, no names were used in the transcript. Each participant received his or her own code, making it visible which recordings belong to the same professional. The codes of each participant were kept confidential and were only accessible to two researchers (RR and DdR). Two researchers (RR and DdR) performed the data coding and analysis of the consensus meeting independently. The researchers discussed dissimilarities until consensus was reached. After the analysis of the international consensus round, an official translator reviewed the competency profile on use of language.

The study protocol (LCI-413) was reviewed by the Clinical Expertise Centre of the National Institute for Public Health and the Environment. Based on this review, they determined that the research plan did not fall under the scope of the Dutch law on medical research involving humans. All necessary precautions were taken to protect the anonymity and confidentially of the participants; in the invitation letter, participants were informed about their voluntary participation and informed that they were free to decline at any time. Furthermore, the participants were informed that their responses were processed anonymously and only used for research purposes.

# Results

## Literature search and extraction of competencies

The literature search for competency profiles resulted in 23 unique studies included in title- and abstract screening, and 7 studies included in the full-text screening, of which two had the highest applicability and were therefore selected. These profiles were ASPHER's European List of Core Competences for the Public Health Professional [28] and Public health emergency preparedness–core competencies for the EU Member States [40].

Table 1. Classification of competencies based on median scores and levels of agreement.

| Median Rating Level of agreement | ≥ 7 | <7 |
|---|---|---|
| Agreement: ≥ 70% within the 7–9 range | Relevant | - |
| Agreement: <70% within the 7–9 range | Uncertain | Irrelevant |
| Scores spread over the entire 1–9 range | Uncertain | Uncertain |

The literature search for airport guidelines resulted in 29 unique documents. Nine were included during the full text screening. Three guidelines had the highest applicability and were therefore selected, being The Handbook for the Management of Public Health Events in Air Transport [29], Coordinated Public Health Surveillance between Points of Entry and National Health Surveillance Systems [41] and The Guide for Public Health Emergency Contingency Planning at Designated Points of Entry [42].

During data extraction, fifteen tasks and 255 competencies were selected from the source documents. After data reduction, around 110 competencies and exactly eleven tasks were left. Aggregating double competencies resulted in 34 competencies which originated from existing competency profiles and 27 from reformulated textual fragments from the air transport guidelines. As a result, the concept competency profile consisted of 61 competencies divided over eleven tasks (c = the number of competencies): Communication (c = 2), Collaboration (c = 3), Professionalism (c = 3), Training (c = 4), Contingency planning (c = 8), Surveillance (c = 8), Risk assessment (c = 7) and Outbreak investigation (c = 6), Management of ill and exposed travelers (c = 6), public health measures (c = 9), and evaluation and recovery (c = 5). **Fig 2** displays all tasks.

## Sample

Thirty professionals were approached for the national study by e-mail. Ten were approached personally; professionals from the AFO and AMS were approached as a group (n = 20). Professionals from PHS, IMA, AMS, AAS, and DMO agreed to participate, professionals from ATC, AFO, and FMA could not participate in the study due to time constraints. However, one participant from AAS had previously worked at the AFO and voluntarily stated as a note in the questionnaire to keep this experience with AFO in mind while filling in the questionnaire. Nine out of ten included participants that completed the questionnaire had extensive experience in infectious disease preparedness and response at Schiphol Airport. The input from the DMO participant was not included in the analyses since she pointed out in an e-mail to the researchers that she is solely involved in the organization of training courses and therefore unable to grade the competencies in terms of relevance.

Sixteen professionals from thirteen countries were approached for the international study, of which ten from ten countries (Austria, Cyprus, Germany, Italy, Ireland, Malta, Slovenia, Spain, Switzerland, and Poland) accepted to participate and completed the questionnaire. Nine professionals (from Austria, Cyprus, Germany, Italy, Ireland, Slovenia, Spain, Switzerland, and Poland) attended the consensus meeting. Their self-assessed competence areas were equally represented among professionals with most representing more than one area: six from the ten professionals were involved in preparedness planning, six in decision making and six in implementation of measures at the airport level. On the national level, five of ten participants were involved with preparedness planning. Table 2 shows the demographic characteristics of the included participants.

## National study

The first questionnaire round resulted in 40 "accepted" competencies, 3 "irrelevant" competencies, and 18 competencies that were marked as "uncertain" (**Fig 3**). Participants provided three main reasons for their low grading of relevance: (1) not relevant to all professionals; (2) concerning individual risks and not aimed at public health; (3) out of scope, primarily focused on other areas (e.g., laboratories). The inconsistency in relevance scores of uncertain competencies reflected different professions. The participants proposed 23 additional competencies. Three participants confirmed, five partly, and one denied that the competency profile reflected

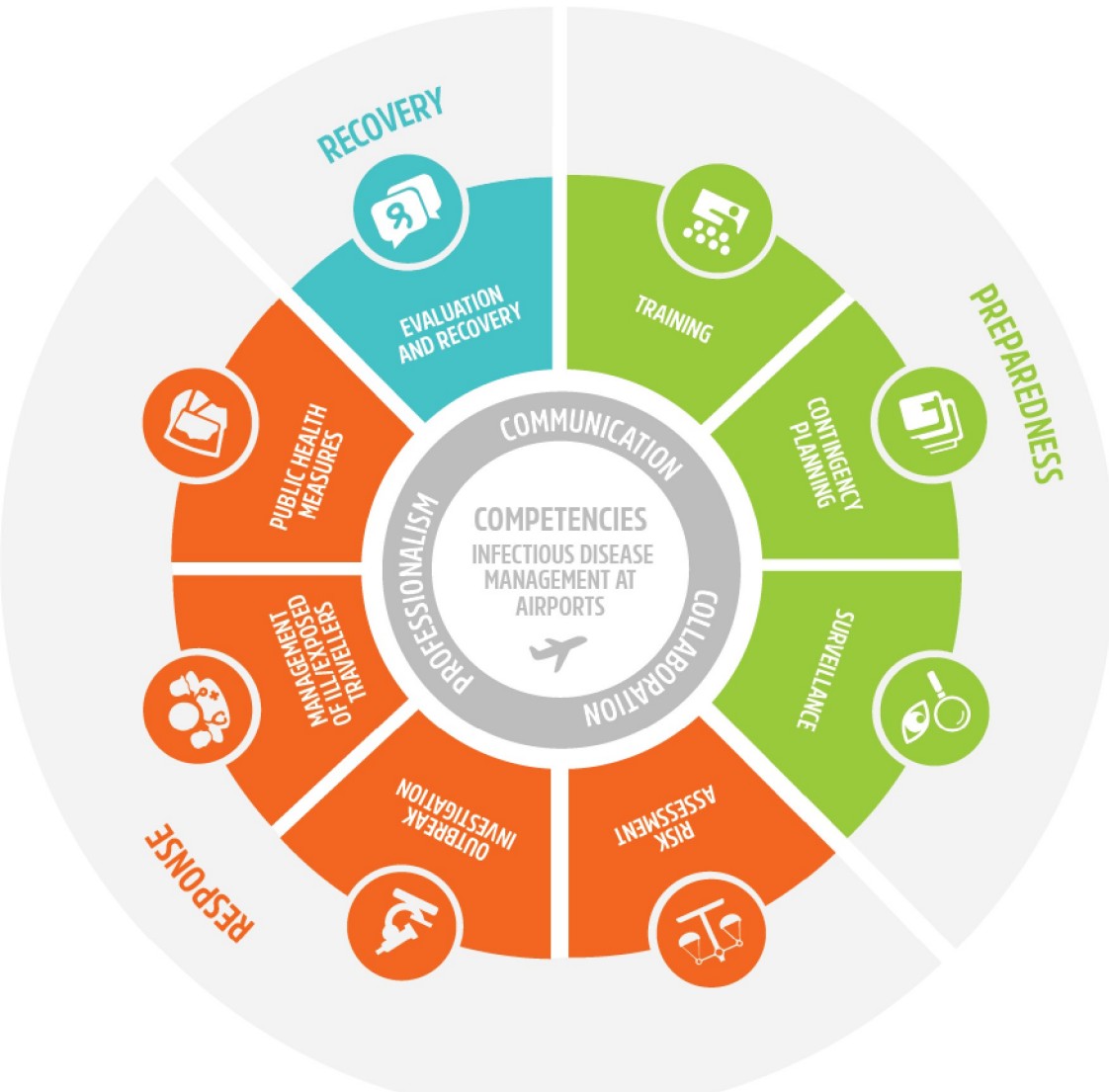

**Fig 2. Framework of the competency profile for infectious disease management at airports.** Own compilation.

their competencies in infectious disease preparedness and response at airports. No irregularities were found within the data analysis.

Five of nine participants attended the consensus meeting. The participants from DMO, AMS, and one from AAS could not attend the meeting due to time constraints. The"uncertain" competencies were discussed of which three were excluded, ten were included after textual amendment, and five were included without textual amendment. Textual amendments included adjusting the target group described within the competency or adapting a verb from an executive to an advisory focus. Of the 23 suggestions for additional competencies, three were included. These included competencies entailed the understanding of the logistical structure and functioning of airports, implementing contact tracing, and contacting professionals with extensive epidemiological knowledge for outbreak investigation. Ten competencies that were already included were reformulated. The group reached consensus for all competencies, no voting rounds were initiated. The reflection after the consensus meeting resulted in several

**Table 2. Demographic characteristics of participants.**

| Demographic characteristic | National study | | International study | |
|---|---|---|---|---|
| | Digital questionnaire | Consensus meeting | Digital questionnaire | Consensus meeting |
| **Participants (n)** | 9 | 5 | 10 | 9 |
| **Sex (n)** | | | | |
| Male | 5 | 2 | 6 | 5 |
| Female | 4 | 3 | 4 | 4 |
| **Profession (n)** | | | | |
| Community Medicine and Infectious Disease Control (PHS) | 2 | 2 | | |
| International Medical Advice (IMA) | 2 | 2 | | |
| Airport Medical Services (AMS) | 2 | 0 | | |
| Continuity & Crisis management (AAS) | 2 | 1 | | |
| Disaster Medicine Organization (DMO) | 1 | 0 | | |
| **Functional level (n)** | | | | |
| Airport level–PH Decision making | | | 6 | 5 |
| Airport level–PH Preparedness planning | | | 6 | 6 |
| Airport level–PH Measure implementation | | | 6 | 5 |
| National level–PH Preparedness planning | | | 5 | 5 |
| **Experience in current profession (n)** | | | | |
| <5 years | 1 | 1 | 2 | 2 |
| 5–15 years | 5 | 2 | 4 | 4 |
| >15 years | 3 | 2 | 4 | 3 |
| **Experience with infectious disease management (5-point Likert scale)** | | | | |
| 1 (very inexperienced) | 0 | 0 | 0 | 0 |
| 2 (inexperienced) | 0 | 0 | 0 | 0 |
| 3 (neither inexperienced/experienced) | 3 | 1 | 4 | 4 |
| 4 (experienced) | 5 | 3 | 6 | 5 |
| 5 (very experienced) | 1 | 1 | 0 | 0 |

N = number of participants.

strengths and difficulties of the profile. During the national study, participants stated that they had enjoyed discussing the profile and this had improved their understanding of roles and responsibilities of different professions involved.

Several participants involved in design and organization of training and exercises would use the profile either to derive training goals or as a background document for participants. Suggestions for improvements were to design a better manageable length or more practical format of the profile, either by splitting it for different disciplines, or connecting competencies to specific tasks. One participant indicated that the content of the competencies had become more evident after the meeting (**Table 3**).

## International study

The questionnaire round with the international panel resulted in 51 "accepted" competencies, seven that were marked as "uncertain", and one "irrelevant" competency. The excluded competency entailed balancing costs and results during PH response. The inconsistency in relevance scores of uncertain competencies reflected different professions. The participants proposed twelve additional competencies. Seven participants confirmed, and three partly confirmed that the competency profile reflected their competencies in infectious disease

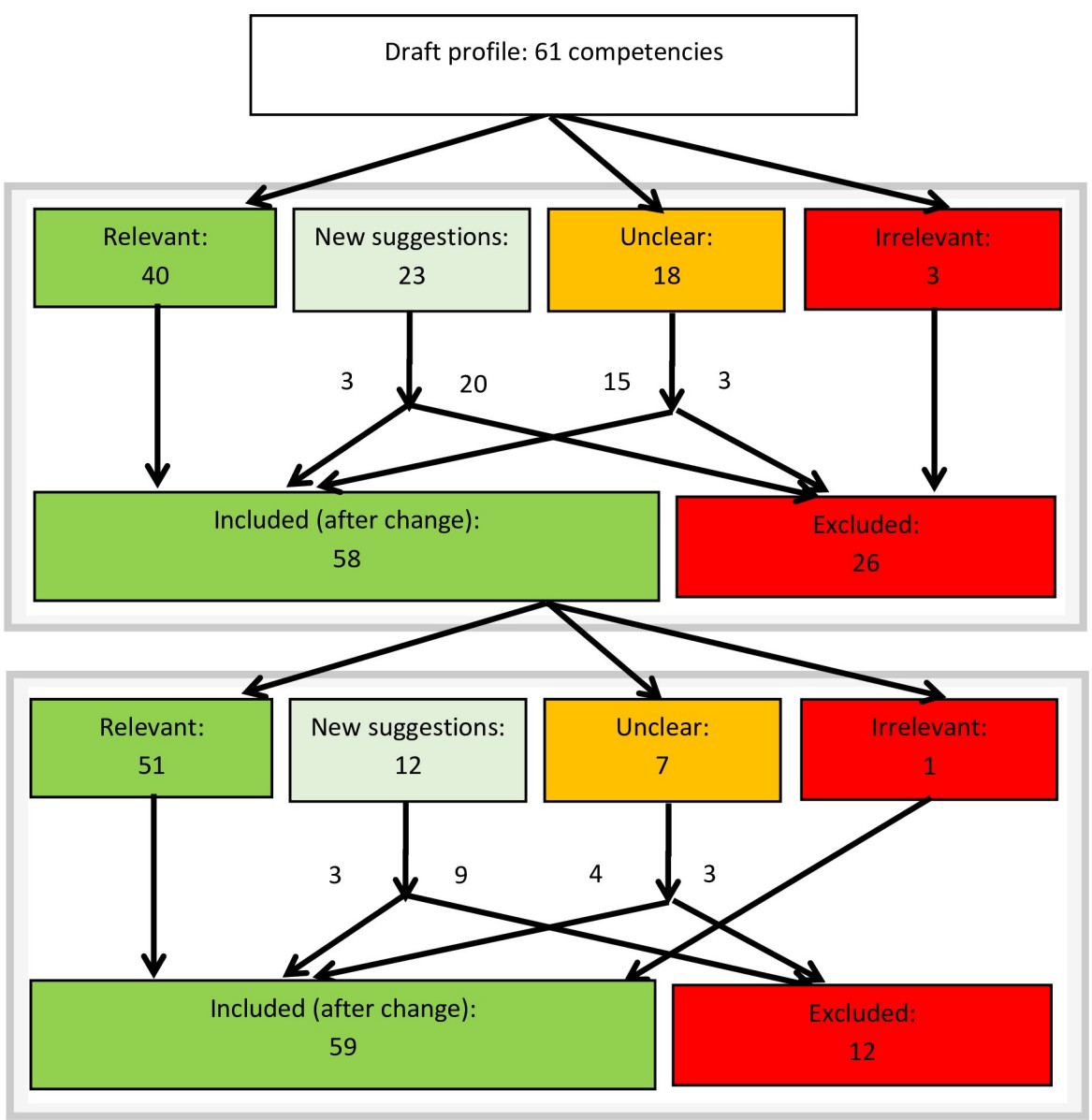

**Fig 3. Flow of competencies during the national and international rounds.**

preparedness and response at airports. No irregularities were found within the data analysis. Six of the ten participants stated to have imagined a case during completion of the questionnaire, of which all referred to EVD or viral hemorrhagic fevers in general. Other named diseases were seasonal influenza, tuberculosis and measles.

Nine out of ten participants attended the consensus meeting. During discussion of the first competency, the group initiated a voting procedure which was used during the majority of competencies. The "uncertain" competencies were discussed, of which three competencies for surveillance and risk assessment were excluded because they were not considered as tasks at the local level. Three competencies were included without textual amendments, and one after further specification of the target group. The only excluded competency in the questionnaire round- on taking the costs in consideration during PH response—was decided yet to be

**Table 3. Quotes of consensus meeting participants regarding the usability level of the refined competency profile.**

| Participant | Corresponding quotes |
|---|---|
| 1) Airport level, PHA, nurse, f, national panel | "I think it is very useful for setting up exercises and setting goals." (translated from Dutch) |
| 2) Airport level, IMA, MD, f, national panel | "When filling in [the questionnaire], it was so abstract and I could not translate it into practice. Now, I think it is a very useful list, certainly now that we have discussed it with different perspectives."(translated from Dutch) |
| 3) National level, MD, m, international panel | "I would use it as a source of competencies that you like to address in a training or exercise or anything like that. So, if you are running a table top or live exercise at the airport, these are the aspirational competencies that you would like, not only for the public health staff but for the whole response." |
| 4) National level, epidemiologist, f, international panel | "I really like it actually that you make this profile especially for the Point of Entry. I have never seen anything like this before but it is really nice. I consider it to be a good reflection of someone being responsible for working directly and being involved in public health measures at the airport." |

PHA = public health authority at the airport; IMA = international medical advice; MD = medical doctor; f = female; m = male.

included. Of the twelve suggestions for additional competencies, four were added to other existing competencies, five were declined, and three were included. These included competencies entailed knowledge of specific terminology in use, data management, and network management of key partners to assure rapid response and recovery.

Analysis of the discussions during and after the consensus meeting indicated that people had sometimes doubted the relevancy of certain competencies due to difference between their own functional level and the scope of the profile (airport level). Also, the differences between several countries in the division of tasks and responsibilities became clear. Participants would use this profile as a source for training goals at the airport or as a background document during training (**Table 3**). To enhance the profile in any kind, they suggested to use the profile during a future training or exercise or to discuss it with their entire team at a local airport.

The final competency profile (**Table 4**) consisted of 59 competencies that were categorized into eleven tasks: Communication ($c = 3$), Collaboration ($c = 4$), Professionalism ($c = 2$), Training ($c = 3$), Contingency planning ($c = 9$), Surveillance ($c = 5$), Risk assessment ($c = 6$), Outbreak investigation ($c = 8$), Management of ill / exposed travelers ($c = 6$), Public health measures ($c = 9$), and Evaluation and recovery ($c = 4$). The majority of competencies entailed the roles of the Health Expert ($c = 26$) or Organizer ($c = 20$). Four competencies involved the scholar who is needed during outbreak investigation and implementation of health measures.

## Discussion

In this study, we developed and validated a competency profile for professionals involved in infectious disease preparedness and response at airports. To the best of our knowledge, this is the first systematically developed competency profile describing the competencies for the air travel environment. The multidisciplinary consensus procedure ensures coverage of all major aspects of preparedness, response and recovery; the international consensus procedure with experts from various countries provides content and face validity to the competency profile and increases its potential for international application.

A setting-specific profile is required because of the specific requirements according to international regulations [9], and the numerous actors which are involved from a range of sectors

**Table 4. Competency profile for professionals involved in infectious disease preparedness and response at airports.**

| | |
|---|---|
| **General** | |
| **Communication** | |
| Communicator | • Understand and implement the basic principles of risk communication to airport and airline staff, travelers, the public and media.<br>• Establish trust with airport and airline staff, travelers, the public and media by using rapid communication channels and ongoing two-way communication.<br>• Understand the terminology used by different levels and organizations at the airport. |
| **Professionalism** | |
| Professional | • Minimize the discomfort or distress associated with public health measures experienced by crewmembers, ground staff, and passengers.<br>• Apply relevant laws to data collection, storage, management, dissemination and use of information. |
| **Collaboration** | |
| Collaborator | • Understand the importance of multidisciplinary collaboration during acute outbreak management.<br>• Be an effective team member, adopting the role necessary to contribute constructively to the accomplishment of tasks by the group.<br>• Participate in the implementation of established plans which ensure the continuity of operations.<br>• Create and manage a network of key partners in rapid response and recovery. |
| **Preparedness** | |
| **Training** | |
| Health expert | • Provide training and exercises on communication within, and between, involved airport organizations and include healthcare providers in this training. |
| Organizer | • Identify training needs, and plan and organize courses.<br>• Periodically practice and test the ability to make decisions in unpredictable circumstances. |
| **Contingency planning** | |
| Health Expert | • Be familiar with job-related standards and recommended practices concerning infectious disease control of national and international aviation organizations (IATA, ICAO and CAPSCA).<br>• Periodically assess whether the implementation of strategies, standard operating procedures (SOPs) and action plans requires any changes.<br>• Before the response operation, identify which triggers will require key decisions to be made during the outbreak response (keeping in mind that triggers may need modification to fit specific situations).<br>• Before the response operation, plan for the storage and stockpiling of medical and non-medical countermeasures. |
| Organizer | • Understand the logistical structure of the airport and the international context of airports and their functioning.<br>• Identify key partners and develop a common understanding of roles, resources, planning assumptions, risks or vulnerabilities, and information needs.<br>• Support core-capacity-building at the airport and understand the importance of it.<br>• Develop, test and evaluate a Public Health Emergency Contingency Plan (PHECP) on a periodical basis.<br>• Provide healthcare workers with clinical guidelines for emerging infections from abroad, especially those that may be carried by travelers and the severely contagious. |
| **Surveillance** | |
| Health Expert | • Recognize a potentially infectious disease by key symptoms and signs of events among travelers.<br>• Understand the relevance of early detection of public health threats.<br>• Understand the components of surveillance systems and how these work. |
| Organizer | • Understand the roles and responsibilities of local, national and international organizations involved in infectious disease control.<br>• Be familiar with laws on the surveillance and reporting of infectious diseases at national, European Union level and globally (International Health Regulations). |
| **Response** | |
| **Risk assessment** | |

(*Continued*)

**Table 4.** (Continued)

| | |
|---|---|
| Health Expert | • Understand risk analysis frameworks, with the elements of risk assessment, risk management and risk communication.<br>• Determine when a risk assessment should be carried out, and appropriate measures taken.<br>• Perform a risk assessment and continuously review the risk assessment as further information becomes available.<br>• Interpret the diagnostic and epidemiological significance of laboratory tests reports. |
| Organizer | • Collect and integrate the facts of an event, based on information from multiple sources, including the traveler, the aircraft operator, ground-based medical services for aircraft in flight (when available) or the agent responsible for the baggage or cargo.<br>• Know when case reports or clusters require further investigation, and how to initiate such investigations. |
| **Outbreak investigation** | |
| Health Expert | • Conduct outbreak investigations to identify pathogens and other agents, characterize affected population groups, and sources of exposure.<br>• Use reliable systems for disseminating case definitions to standardize both the diagnosis and the reporting of case numbers (e.g. confirmed, suspected, probable, or possible, cases).<br>• Systematically generate required information about the number of travelers such as those targeted for screening, screened, referred to secondary screening, and identified as confirmed cases.<br>• Implement contact-tracing based on a careful, case-by-case, risk assessment basis, taking into account factors such as feasibility, the severity of the disease and its potential for epidemic spread, the infectivity of index patients, and the duration of the trip. |
| Organizer | • Identify who is responsible at national level for receiving the information on the investigation from the local or intermediate level health authority. |
| Scholar | • Maintain up-to-date and job-specific knowledge about characteristics of infectious diseases such as the reservoir, potential sources, modes of transmission, risk groups, and duration.<br>• Be able to contact professionals who have the biological, clinical, and epidemiological knowledge necessary to characterize (potentially novel) pathogens and other agents responsible for an outbreak disease.<br>• Use evidence-based methods to identify and recommend control and preventive measures to control an outbreak. |
| **Management of ill and exposed travelers** | |
| Health Expert | • Provide ground-based medical support (GBMS) regarding infectious disease events, including medical recommendations to manage the discovery of a suspected communicable disease during flights, to support decisions regarding medical treatment and use of on-board medications or equipment.<br>• Assess the health status and travel history of travelers arriving from, or going to, an affected region, or who have been exposed to a potential public health risk during air travel. |
| Organizer | • Provide disembarking travelers with information regarding the precautions to take in the event of illness, information sources for any updates on the event and the public health authority (PHA) contact information where subsequent enquiries can be made.<br>• Provide advice concerning the appropriate parking stand for an incoming affected aircraft and the order of disembarkation of passengers.<br>• Provide advice to ensure port health staff respond efficiently so as reduce the time that travelers spend on a board-affected aircraft, and identify space requirements for interviews and health assessments of arriving travelers.<br>• Provide advice on a traveler's possible transfer to a medical facility by ambulance and facilitate the rapid transport of suspected cases of an infectious disease. |
| **Public health measures** | |
| Health Expert | • Recognize when it is necessary to wear Personal Protective Equipment (PPE), what PPE is required, where the equipment is stored and how PPE is donned or doffed.• Determine triggers for appropriate public health measures to be taken, such as travel restrictions, quarantine, treatment and isolation, that are commensurate with the risk and do not unduly interfere with international travel.• Relate information regarding medical clearance for travelers with health conditions which may affect their suitability for air travel.<br>• Provide information regarding vaccination or other prophylaxis to affected travelers.<br>• Determine, based on results of the inspection, if further disinfection, decontamination, disinsection or deratting measures of the aircraft or at the airport are required.<br>• Recognize when to implement the special handling of baggage or cargo from affected regions, including inspection, fumigation, and other decontamination of possibly destruction. |

(*Continued*)

**Table 4.** (Continued)

| Organizer | • Assess whether the costs of the public health measures and resulting liabilities are proportionate to the risk.<br>• Equip relevant airport and airline staff with information regarding the public health event so that they can protect themselves and safeguard healthy travelers as required. |
|---|---|
| Scholar | • Organize the use of public health measures underpinned by scientific evidence and expert public health opinions, so as to avoid any contradictory or unnecessary restrictions of individuals. |
| **Recovery** | |
| **Evaluation and recovery** | |
| Health Expert | • Clearly define goals and objectives of the evaluation of training, exercises or real response.<br>• Develop a formal evaluation of the response, including recommendations for prevention and mitigation for future incidents, and share the evaluation with all stakeholders when the public health event is under control or has concluded. |
| Organizer | • Deactivate the plan and return to recovery once the situation is under control or able to be de-escalated.<br>• Update plans according to the key lessons learnt after a formal review of training, exercises or real response. |

other than public health [43]. As such, airports function as coherent subsystems within the broader public health system. The ASPHER's European List of Core Competences, which we used as a main source, acknowledges the challenge to apply their general public health competency profile to such systems [28]. However, the use of competencies can be pivotal in targeted and effective training for professionals, and the evaluation of their competence, as is proven by many studies [18, 36, 44–46]. To support the development of well-functioning infectious disease systems at airports, we made general formulations more specific and added setting-specific knowledge, skills and attitudes to end up with this airport setting-specific competency profile. We, therefore, demonstrate in this study, that it is possible to design a competency profile for airports across countries.

Presumably, other points of entry, such as ground-crossings or ports, would also benefit from a setting-specific profile. These have a regulated and sustainable role in prevention of cross-border spreading of diseases and face comparable challenges to airports. Examples are fisheries [47], drilling platforms[43], and many other mass gatherings[48] or other places imaginable where an international transfer of people and goods take place, and where infectious disease diseases might be introduced.

During the development of the profile, we learned more about the possibilities and challenges of competencies. Both during the extraction of competencies from the literature, as well as during the consensus meeting we experienced that the concise and theoretical formulations of the competencies sometimes thwart their usability. It happened to us, as well as to the participants, that competencies had to be read and reread out loud before the full meaning of a competency was fully understood. The usability of the competency profile can be enhanced in several ways. For example, the use of entrustable professional activities (EPA), as used in medical education, may bridge a potential gap between the theory of competencies and practice. An EPA tells whether a professional can be entrusted to carry out all critical activities of a major task [19]. They integrate the demonstration of competence with the respective supervision level and hereby form a usable tool during education or training activities.

Because our profile covers competencies required by the team of professionals involved in infectious disease management, participants experienced that competencies could only be translated into practice in close collaboration with different disciplines replicating findings of previous studies [49]. Another possibility to increase the usability of this profile is to merge disciplinary competencies into functional roles at an airport. In line with recommendations from

this study's participants, the profile could be minimized to a specific group of people, for example, first responders. A third option, is to use the profile and apply it in practice. Discussing the profile enhanced usability according to our participants. In addition, we lowered the barrier of translating the theoretically formulated competencies into practice by inviting our participants to think of a possible event.

During the international questionnaire, people noted which event they had used during completion. Remarkable is that all the events listed comprised a viral hemorrhagic fever. While the chance for an event or suspicion of a viral hemorrhagic fever is still low, it is an interesting finding that its high impact, possibly due to the EVD pandemic of 2014–16 and the recent endemic state in the Democratic Republic of the Congo, affects the thoughts of preparations in Europe.

Remarkable findings in the profile are the low number of competencies for recovery and evaluation after an event. In the airport guidelines that we used as a source, evaluation and recovery are scarcely named and hardly elaborated on [29,41,42]. However, the use of after action reviews are a required action according to the IHR, and major benefits of either training or real response are made here [50, 51]. It is therefore worrisome that this phase of the event is hardly elaborated upon in current landmark guidelines and competency profiles, and, consequently, is scored as a rather small topic in this study's competency profile. We cannot determine, however, whether the small number of competencies resembles the real need or resembles a lack of awareness. We suggest to critically review the competencies required for recovery and the after action review after a major event.

It may seem remarkable that many competencies in our profile are knowledge or skills in comparison to attitudes. This could be the result of several aspects. First, we build upon former general profiles and focused on competencies specific for the airport setting. Attitudes such as leadership, flexibility, team work or reflection are mostly general, i.e. not airport-specific. In addition, these are already widely covered in the Team Resource Management skill set, which is additional to our profile and focusses on attitude [36]. However, another possibility is that knowledge and skills are more concrete while attitudes remain harder to distinguish by our participants. The use of this profile in practice should indicate whether attitudes are sufficiently covered.

We consider it as a unique opportunity to thoroughly and extensively develop a competency profile by performing two consecutive RAND modified Delphi studies, one at Schiphol Airport, one of Europe's largest airports, and one internationally with experts from ten European countries. As in every study, we faced several challenges that we tried to cope with. First, composing a draft profile demanded a pragmatic step of combining and formulating competencies of different profiles and guidelines. Diminishing the large amount of overlapping competencies with slight differences in word choice, specificity level and combinations of activities was done as systematically as possible based on thoroughly produced profiles in the international literature to present an assessable profile to the participants. Another challenge was to compose expert panels with optimal representation of all involved disciplines. In the national study, we could not include all professionals involved in infectious disease management at Schiphol Airport. However, we assume to have caught the major perspectives, since professionals that are involved with most tasks were represented. In addition, all included participants were highly experienced in outbreak management and collaborated with the missing experts in daily practice. During the international study, we had to select participants based on their self-assessed competence regarding the subject and could not prevent a mixed group regarding experience in real practice. We therefore consider it a strength that the international participants all participated in the European training course, leading to an equally educated group in the consensus meeting. A third challenge is to design a profile that is internationally

usable while it turned out that designation of tasks between national, regional or airport levels differs among countries. However, we explicitly demanded participants to look further than the specific situation at their airport, leading to a profile that forms a thorough base for training in European countries.

Future steps would be to test the usability and implementation of this profile in real practice by means of trainings, exercises and evaluations. It is our aim that this profile is applied at airports worldwide to facilitate a competent workforce, by integrating it in training or exercising schedules as training material or background reference for organizers. Standardized and extensive use of this profile could help to standardize terminology among professionals, and contribute to better communication and coordination. Subsequent development of tools such as an EPA profile on a European level, or the implementation of competencies into discipline specific profiles at specific airports are possibilities. Finally, we hope that this competency profile can be used as a basis to develop specific competency profiles for points of entry other than airports, or other settings important in cross-border spreading of disease.

In this study, we developed an interprofessional competency profile for professionals involved in infectious disease preparedness and response at airports by means of landmark literature and expertise from professionals in eleven European countries. This profile could be considered as promising contribution to improving preparedness and response to cross-border spreading of infectious diseases and future training and research this direction.

## Supporting information

**S1 File. Literature searches.**
(DOC)

**S2 File. Full list of competencies extracted from literature.**
(DOC)

**S3 File. Reduced list of extracted competencies from literature.**
(DOC)

**S1 Data. Overview of analyses and profile development.**
(XLSX)

**S1 Questionnaire. Questionnaire national data collection.**
(PDF)

**S2 Questionnaire. Questionnaire international data collection.**
(PDF)

## Acknowledgments

The authors are thankful to all participants for sharing their knowledge and practical experience. We thank Saskia van Egmond for her help in selecting the best representable panel, and for her advice and enthusiasm during the project. We thank Eline Westland—Hogenbirk for the configuration of Fig 2; Jeannette de Boer for her reflections, including on the future applications and usability of our profile; and Corien Swaan for her contributions before and during the international consensus meeting; and our colleagues from the EU Joint Action Healthy Gateways for the opportunity to perform our study during the training-of-trainers in Belgrade.

## Author Contributions

**Conceptualization:** Doret de Rooij, Evelien Belfroid, Aura Timen.

**Data curation:** Doret de Rooij, Rebekka Rebel.

**Formal analysis:** Doret de Rooij, Rebekka Rebel, Aura Timen.

**Investigation:** Doret de Rooij, Rebekka Rebel, Evelien Belfroid.

**Methodology:** Doret de Rooij, Rebekka Rebel, Evelien Belfroid, Aura Timen.

**Resources:** Christos Hadjichristodoulou, Aura Timen.

**Supervision:** Doret de Rooij, Jörg Raab, Evelien Belfroid, Aura Timen.

**Validation:** Doret de Rooij, Jörg Raab.

**Visualization:** Rebekka Rebel.

**Writing – original draft:** Doret de Rooij, Rebekka Rebel.

**Writing – review & editing:** Doret de Rooij, Jörg Raab, Christos Hadjichristodoulou, Evelien Belfroid, Aura Timen.

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
