## [Decision Letter · Decision Letter 0]

23 Mar 2020

PONE-D-20-01304

Development of a competency profile for professionals involved in infectious disease preparedness and response in the air transport public health sector

PLOS ONE

Dear Mrs de Rooij,

Thank you for submitting your manuscript to PLOS ONE. After careful consideration, we feel that it has merit but does not fully meet PLOS ONE’s publication criteria as it currently stands. Therefore, we invite you to submit a revised version of the manuscript that addresses the points raised during the review process.

Please see details in Additional Editor Comments as well as Reviewer Comments.

We would appreciate receiving your revised manuscript by May 07 2020 11:59PM. To enhance the reproducibility of your results, we recommend that if applicable you deposit your laboratory protocols in protocols.io, where a protocol can be assigned its own identifier (DOI) such that it can be cited independently in the future. For instructions see: http://journals.plos.org/plosone/s/submission-guidelines#loc-laboratory-protocols

We look forward to receiving your revised manuscript.

Kind regards,

Michio Murakami

Academic Editor

PLOS ONE

Additional Editor Comments (if provided):

Please consider following points.

1. The description of "n" is confusing, because some represent the number of participants, other mean the number of competencies. Please avoid the confusion due to this description.

2. The authors need to add more explanations regarding literature review. How did the author perform literature search to finally find two landmark profiles and three guidelines? More details (e.g., keywords, language, criteria) are necessary.

3. Did the authors confirm the quality of the sentences translated from Dutch to English? Typically, this may need two points. First, the sentences translated to English should be checked by a native speaker. Second, the authors should check sentences that were "back-translated" from English to Dutch again. Did the authors perform to confirm these two points?

Journal Requirements:

Please ensure that your manuscript meets PLOS ONE's style requirements, including those for file naming. The PLOS ONE style templates can be found at http://www.plosone.org/attachments/PLOSOne_formatting_sample_main_body.pdf and http://www.plosone.org/attachments/PLOSOne_formatting_sample_title_authors_affiliations.pdf

Reviewers' comments:

Reviewer's Responses to Questions

**Comments to the Author**

1. Is the manuscript technically sound, and do the data support the conclusions?

Reviewer #1: Yes

Reviewer #2: Yes

2. Has the statistical analysis been performed appropriately and rigorously? 

Reviewer #1: N/A

Reviewer #2: Yes

3. Have the authors made all data underlying the findings in their manuscript fully available?

Reviewer #1: Yes

Reviewer #2: Yes

4. Is the manuscript presented in an intelligible fashion and written in standard English?

Reviewer #1: Yes

Reviewer #2: Yes

5. Review Comments to the Author

Reviewer #1: This is quite a meaningful study trying to identify the competences for emergency preparedness and response of an airport. The operation of an international airport is supported by many units, private and public. So is the emergency management. How the works and responsibilities are arranged for all units working at the airport is complicated and varied among countries. Regarding to infectious diseases emergency management, it might not be just confined at the airport and will extend further into the community or neighboring area, and even countries.

Since the management of infectious diseases emergency at the airport might involve authorities and units outside, the first thing needs to do when discussing about the competences of the airport is to identify what exactly the airport needs to do and its responsibility. Then it will be appropriate to match needed competences for their jobs.

The questionnaire used in national panel was in Dutch and translated into English for International validation. Will there be any worry that the translation resulted in semantic differences or gaps?

Whether deletion of those items being recognized as irrelevant affects the implementation of the IHR core capacity? I think it is an important thing to reassure.

Reviewer #2: The study describes the process leading to developing competences for professional at Schiphol Airport in a mixed method of qualitative and quantitative study.

The methods section is very well described and the combination of the technics (Delphi technique, literature review, desk review of documents and interviews) technically sound.

The following are issues to be addressed by the authors:

In the section: Study population and recruitment

1- “The research team aimed for two panels, of which each consisted out of 8 – 15 participants”,

Not clear the exact number for each of the panels, unless the number was not fixed. In this case, it should state in the manuscript.

2- “The international panel was recruited by selecting professionals who participated in a European, face-to face 3-day training course on infectious disease control at designated airports”

I am not clear why for international panel it has been used a different criterion from the national panel. I understand the selected people who were trained, but it doesn’t mean they are experienced people. And a “self-declared competence on local and national level before the training to approach a variety of professionals” also does not necessarily means they have enough experience to decide for competence for an entire country.

3- In the section: Design, Step 1 – Literature search and extraction of competencies

The literature search requires more details. What were the inclusion and exclusion criteria? How many studies were finally reviewed, etc.

4- There is a: “Table 1. Classification of competencies based on median scores and levels of agreement.” What is the “n=?” in this table?

Results

5- “Thirty professionals were approached for the national study by e-mail. Ten were approached personally; professionals from the AFO and AMS were approached as a group (n=20)”

I think these details should be in the methods section. Leaving for authors and editors to decide

6. PLOS authors have the option to publish the peer review history of their article (what does this mean?). If published, this will include your full peer review and any attached files.

Reviewer #1: Yes: Chang-Hsun CHEN

Reviewer #2: Yes: Jeremias Domingos Naiene

---

## [Author Response · Author response to Decision Letter 0]

29 Apr 2020

Bilthoven, 29 April 2020

Distinguished Editor, Distinguished reviewers, 

Thank you for sharing the results of the review of our manuscript entitled “Development of a competency profile for professionals involved in infectious disease preparedness and response in the air transport public health sector”. We are very pleased to read that the reviewers believe that this subject is important and we are very grateful for the thorough review process. The feedback and suggestions enabled us to further improve the manuscript. 

Below you will find all comments made by the reviewers and editor, followed by our adjustments in the manuscript and further clarifications. We hope we have been able to address your and the reviewers’ concerns.

Response to the comments made by the Editor:

• “1. The description of "n" is confusing, because some represent the number of participants, other mean the number of competencies. Please avoid the confusion due to this description.”

We understand the confusing effect of using ‘n’ for both the number of participants and the number of competencies. Therefore, we referred to the number of competencies by using ‘c’. Now, the ‘n’ always refers to the number of participants. 

• “2. The authors need to add more explanations regarding literature review. How did the author perform literature search to finally find two landmark profiles and three guidelines? More details (e.g., keywords, language, criteria) are necessary.”

We agree that this part of the methodology is least elaborated on. The main reason for this literature search was to find reliable documents to use in our study. We therefore did a scoping search aimed at the identification of important competency profiles and airport guidelines to be able to combine them. We acknowledge that the description of the search requires further attention. Therefore, in the methods section we added for both the competency profiles and the airport guidelines the search terms, the screening criteria for title- and abstract screening, for full-text screening, and for selection. In the results section of the manuscript, we added for both the competency profiles and airport guidelines the number of documents selected by title- and abstract screening and full-text screening. In addition, an overview of the full search, the screening process, the results of the title/abstract and full-text screening is added as an additional Supporting Material File ‘S1. Literature searches’. In this way, we trust to provide readers of the manuscript with sufficient information regarding the literature review that we performed as step 1 of this study. 

• “3. Did the authors confirm the quality of the sentences translated from Dutch to English? Typically, this may need two points. First, the sentences translated to English should be checked by a native speaker. Second, the authors should check sentences that were "back-translated" from English to Dutch again. Did the authors perform to confirm these two points?“

The first, national round of this Delphi study was performed in Dutch regarding information delivery and discussion. But all competencies that were scored, were already formulated in English. And these have remained in English during the whole process. We now state in the manuscript explicitly that “[t]he introduction, privacy statement, and demographic questions were in Dutch, but the questions regarding the relevance of competencies, as well as the competencies were in English.” 

However, a semantic problem might still remain since we as authors have composed several first versions of the competencies and several respondents, many being non-native speakers, made adaptations to the competencies. We trust in the fact that all experts work in international environments, but we considered it of added value to check the competency list with a native speaker. In the previous three weeks (April 2020), we approached an official translator to perform an additional language check of the competency profile, as presented in Table 4 in the manuscript. Based on this translator’s feedback, we made several textual adjustments mainly addressing word order and prepositions. All textual changes are visible in Table 4 of the manuscript version with tracked changes. In the method section, we added a description of the use of this translator in the development of the profile. 

Response to the comments made by Reviewer 1:

• “The questionnaire used in national panel was in Dutch and translated into English for International validation. Will there be any worry that the translation resulted in semantic differences or gaps?”

Thank you for this point. First, we would like to clarify that the questionnaire used in the National panel was in Dutch, but all the competencies were stated in English both during the questionnaire as well as during the consensus meeting. We further clarified this in the manuscript, by stating now that “[t]he introduction, privacy statement, and demographic questions were in Dutch, but the questions regarding the relevance of competencies, as well as the competencies were in English.” 

The only worry that may be remaining is that both in the National and International panel, the majority of participants were non-native English speakers. Although we consider this a general challenge in international verbal knowledge exchange itself, to avoid semantic errors with the readers, we have approached an official translator to check the competency profile as presented in Table 4 of the manuscript on language. Based on this translator’s feedback we made several textual adjustments mainly addressing word order and prepositions. All textual changes are easy traceable in Table 4 of the manuscript version with tracked changes. We also added the translator’s work to the description of the competency profile development in the methods section. 

• “Whether deletion of those items being recognized as irrelevant affects the implementation of the IHR core capacity? I think it is an important thing to reassure.”

Thank you for this note; we regard it of utmost importance and we have thought about this as well. We consider it improbable that the deleted competencies, derived from WHO guidelines, would lead to a decreased implementation of the IHR core capacities. First, the competency level is one of the levels where efforts need to be performed to have sufficient capacity. But also, for example, capacity and a flexible organizational structure are important. Regarding the capacities, these are evaluated on a yearly basis. This study’s competency profile serves training development and evaluation, team formation and so on, but does not replace capacity and several structures that need to be in place. This profile does not relieve countries from their obligation to implement core capacities but is a useful translation of these for the requirements of individuals and teams. 

Response to the comments made by Reviewer 2:

• Study population and recruitment

o “1- “The research team aimed for two panels, of which each consisted out of 8 – 15 participants”, Not clear the exact number for each of the panels, unless the number was not fixed. In this case, it should state in the manuscript.”

Thank you for this note, we understand that this is confusing. The 8-15 participants was something we aimed for based on the RAND Guide for performing Delphi studies (Fitch). However, for clarification, we added into the manuscript the exact numbers of participants that participated in both panels. 

o “2- “The international panel was recruited by selecting professionals who participated in a European, face-to face 3-day training course on infectious disease control at designated airports” I am not clear why for international panel it has been used a different criterion from the national panel. I understand the selected people who were trained, but it doesn’t mean they are experienced people. And a “self-declared competence on local and national level before the training to approach a variety of professionals” also does not necessarily means they have enough experience to decide for competence for an entire country.”

Thank you for your concern. First of all, we had slightly different goals for the national and international panels. The national focusing on a mix of experts with different roles in infectious disease management to cover a broadly accepted profile among professionals at an airport. The international round focused on international acceptance. Therefore, we focused less on the perfect mix of professionals, but on experts from different countries and working levels. 

To further explain the selection of the international panel. This panel was composed based on two factors. The first and main factor was their reputation on the matter, as experts in the field. This assessment could be done well, due to the expertise of several authors in the field, and involvement in the Joint Action Healthy Gateways network in which over 29 countries participate. Indeed, we did this among the participants and trainers of the JA Healthy Gateways training. This training was aimed at so-called ‘designated’ points of entry, that are legally obliged to prepare for infectious disease events. This further guaranteed professionals that are expert on this matter. Based on the reputation of their expertise, we, as authors, made the first selection of experts to be invited. 

Second, in this subgroup of experts, we applied a second criterion, which is the combination of their characteristics such as geographic location within Europe, working-level (national vs. local), and their self-declared domain to have a panel of expertise and reputation across Europe. 

• 3- In the section: Design, Step 1 – Literature search and extraction of competencies

o “The literature search requires more details. What were the inclusion and exclusion criteria? How many studies were finally reviewed, etc.”

Thank you for this point of attention. We briefly elaborated on the search strategy, inclusion criteria and results in the manuscript, but also added the full search, screening process, results of the title/abstract and full-text screening in the supporting material file ‘S1. Literature search’. In this way, we provide readers of the manuscript with sufficient information regarding the literature review that we performed as step 1 of this study. 

o “4- There is a: “Table 1. Classification of competencies based on median scores and levels of agreement.” What is the “n=?” in this table?”

Thank you for this question. Table 1 shows the method that we applied to select the relevant, irrelevant and to be discussed competencies. Because this table is located in the methods section, we do not present any numbers of competencies in this table. 

• Results

o “5- “Thirty professionals were approached for the national study by e-mail. Ten were approached personally; professionals from the AFO and AMS were approached as a group (n=20)”. I think these details should be in the methods section. Leaving for authors and editors to decide.”

Thank you for the advice. We feel your point, but for readability reasons, suggest to leave the outcome of sample selection in the results. 

We are grateful for the thoughtful recommendations and we believe that our revisions have improved the argument and clarity of the manuscript. 

Looking forward to receiving your decision,

Yours sincerely, 

On behalf of all authors,

Doret de Rooij

---

## [Editor Report · Decision Letter 1]

5 May 2020

Development of a competency profile for professionals involved in infectious disease preparedness and response in the air transport public health sector

PONE-D-20-01304R1

Dear Dr. de Rooij,

We are pleased to inform you that your manuscript has been judged scientifically suitable for publication and will be formally accepted for publication once it complies with all outstanding technical requirements.

With kind regards,

Michio Murakami

Academic Editor

PLOS ONE
---

## [Editor Report · Acceptance letter]

7 May 2020

PONE-D-20-01304R1 

Development of a competency profile for professionals involved in infectious disease preparedness and response in the air transport public health sector 

Dear Dr. de Rooij:

I am pleased to inform you that your manuscript has been deemed suitable for publication in PLOS ONE. Congratulations! Your manuscript is now with our production department. 

With kind regards,

on behalf of

Dr. Michio Murakami 

Academic Editor

PLOS ONE